# Low Dose Monacolin K Combined with Coenzyme Q10, Grape Seed, and Olive Leaf Extracts Lowers LDL Cholesterol in Patients with Mild Dyslipidemia: A Multicenter, Randomized Controlled Trial

**DOI:** 10.3390/nu15122682

**Published:** 2023-06-09

**Authors:** Nicholas Angelopoulos, Rodis D. Paparodis, Ioannis Androulakis, Anastasios Boniakos, Georgia Argyrakopoulou, Sarantis Livadas

**Affiliations:** 1Endocrine Unit, Athens Medical Centre, 65403 Athens, Greece; 2Private Practice, Venizelou Str., 65302 Kavala, Greece; 3Center for Diabetes and Endocrine Research, University of Toledo College of Medicine and Life Sciences, Toledo, OH 43614, USA; 4Private Practice, Gerokostopoulou 24, 26221 Patra, Greece; 5Private Practice, Tzanaki Emmanouil 17, 73134 Chania, Greece

**Keywords:** monacolin K, coenzyme Q10, grape seed extract, olive leaf extract, hypercholesterolemia

## Abstract

Certain nutraceuticals, mainly containing red yeast rice, might be considered as an alternative therapy to statins in patients with dyslipidemia, although there is still insufficient evidence available with respect to long-term safety and effectiveness on cardiovascular disease prevention and treatment. The aim of this study was to assess the lipid-lowering activity and safety of a dietary supplement containing a low dose of monacolin K combined with coenzyme Q10, grape seed and olive tree leaf extracts in patients with mild hypercholesterolemia. In total, 105 subjects with mild hypercholesterolemia (low-density lipoprotein cholesterol LDL-C levels 140–180 mg/dL) and low CV risk were randomly assigned into three treatment groups: lifestyle modification (LM), LM plus a low dosage of monacolin K (3 mg), and LM plus a high dosage of monacolin K (10 mg) and treated for 8 weeks. The primary endpoint was the reduction of LDL-C and total cholesterol (TC). LDL-C decreased by 26.46% on average (*p* < 0.001) during treatment with 10 mg of monacolin and by 16.77% on average during treatment with 3 mg of monacolin (*p* < 0.001). We observed a slight but significant reduction of the triglyceride levels only in the high-dose-treated group (mean −4.25%; 95% CI of mean −11.11 to 2.61). No severe adverse events occurred during the study. Our results confirm the LDL-C-lowering properties of monacolin are clinically meaningful even in lower doses of 3 mg/day.

## 1. Introduction

Lowering of the LDL concentration produces benefits in cardiovascular disease prevention equivalent to the percentage of the LDL cholesterol (LDL-C) lowering and the final LDL-C concentration achieved, independent of the modality used to obtain that reduction [1]. Patients with mild hypercholesterolemia (defined as LDL-C levels slightly above the optimal level, related to their individual cardiovascular (CV) risk), might be eligible for some lipid-lowering therapies when nutritional modifications fail to adequately reduce the atherogenic molecules in circulation [2]. This consists of patients requiring lipid-lowering therapy for primary prevention and those with intermediate CV risk, who would be ineligible for statins use due to the risk/benefits ratio, as well as those with statins intolerance or when PCSK9 inhibitors are not available or indicated. These scenarios are of great clinical significance, because it is well known that a large proportion of patients developing CV disease would not fit in the prevention groups or were unable to achieve their primary prevention LDL-C treatment goals with the currently used therapies. This renders very important the development of alternative, mild in regard to side-effects but equally effective, strategies to lower LDL-C in all these populations. Such treatment strategies have been tested worldwide in recent years, in the form of nutraceuticals; the present work aims to assess the efficacy of neurochemical supplements with lipid-lowering effects in terms of LDL-C and CV risk reduction [3].

Red yeast rice (RYR) is a well-known food ingredient in many Asian countries, often utilized in the preparation of various foods, such as fish, meat, and rice wine [4]. It has been used for centuries, starting from the Ming Dynasty, as a remedy to improve digestion and blood circulation [5]. RYR is made through the fermentation of white rice and Monascus purpureus fungus, with the fermentation conditions determining the presence of red dyestuffs, flavors, and other components such as monacolins. Monacolin K is identical to the statin lovastatin (Figure 1) and, thus, inhibits HMG-CoA reductase, the rate-limiting enzyme in cholesterol synthesis [6].

Extracts of RYR, and its chief bioactive compound monacolin K, are currently considered the most efficacious cholesterol-lowering nutraceutical. Previous intervention studies have indicated that RYR containing 5 to 10 mg monacolin K can lower elevated LDL-C levels by 22% to 27% [7,8]. The European Food Safety Authority suggests that a daily intake of 10 mg monacolin K from RYR contributes to maintaining normal LDL-C plasma levels [9]. Studies that utilized RYR products containing a low dose of 3 mg monacolin K in combination with other cholesterol-lowering agents, such as berberine or policosanols, have reported a decrease in LDL-C levels by 20% to 31% [10,11].

The combination of nutraceuticals allows achieving LDL-C reductions by using lower doses of each component and thus reduces the likelihood of adverse events related to a single component [12]. Nonetheless, because these supplements contain additional cholesterol-lowering agents, it is challenging to discern whether the lipid-lowering effect was primarily due to monacolin K or another agent. On the other hand, mixed supplements may have the opposite effect in terms of efficacy and safety. For example, RYR’s potential to inhibit the HMG-CoA reductase enzyme might be neutralized by coupling it with other nutraceuticals, such as plant sterols, to increase the lipid excretion in the bowels [13]. We have recently showed that a combination of RYR containing 10 mg of monacolin with CoQ10, olive leaf extract (OLE), E, grape seed extract (GSE), and vitamin B complex has beneficial effects, significantly decreasing LDL-C concentrations (mean LDL-C reduction 42.35 mg/dL) [14]. However, various studies assessing the effects of RYR food supplements containing monacolins yielded safety concerns when the dose of monacolin reached 10 mg/day [15]. The adverse effects profile was similar to that of lovastatin: (1) musculoskeletal (29.9–37.2% of cases, including 1–5% of rhabdomyolysis), hepatic (9–32%), nervous system (when reported, 12.8–26.9%), gastrointestinal (12–23.1%), and cutaneous and subcutaneous tissues (8–17.3%) [15].

The aim of this study was to assess the lipid-lowering activity and safety of a novel, commercially available dietary supplement containing a low dose of monacolin (3 mg) combined with CoQ10, OLE), GSE, and vitamin B complex in volunteers with moderate elevations in LDL-C concentrations and low cardiovascular risk.

## 2. Materials and Methods

### 2.1. Study Protocol

The present is a prospective, multicentric, open-label, randomized controlled trial conducted at 4 outpatients endocrine clinical centers in Greece, according to a controlled, randomized, and repeated measures design from June 2022 to January 2023 (registration code: ACTRN12623000621617). Patients were eligible to participate in the present study if they were >40 years of age, had mild hypercholesterolemia (fasting LDL-C concentration between 140 and 180 mg/dL), and no indication for statin treatment (10-year atherosclerotic cardiovascular disease risk (ASCVD Risk) <7.5%). The 10-year risk was calculated in all participants using the updated ASCVD Risk Estimator Plus (found in: https://tools.acc.org/ASCVD-Risk-Estimator-Plus/#!/calculate/estimate/ (accessed on 1 June 2022)). Exclusion criteria were the use of lipid-lowering medications, such as statins, within 3 months prior to study enrollment, conditions which produce an increased ASCVD risk, such as diabetes mellitus, known atherosclerotic vascular disease, moderate/severe renal insufficiency (Modification of Diet in Renal Disease calculated glomerular filtration rate, MDRD GFR < 60 mL/min), abnormal liver function tests, excessive alcohol intake, pregnancy, lactation, or the use of oral contraceptives. All patients were encouraged to follow a standardized Mediterranean diet characterized by the high consumption of fish, fruit, vegetables, legumes, olive oil, and unrefined whole grains accompanied by a modest intake of lean meats and alcohol for 8 weeks before study enrollment. After the run-in period, while being on the diet, the lipid profile was assessed, and those who still had elevated LDL-C concentrations and agreed to participate in the study comprised our study population (visit 1-study initiation). A neurochemical compound (Arichol^®^ tablets, Epsilon Health, one tablet every evening after dinner; ingredients are shown in Table 1) with two different doses of monacolin (10 mg and 3 mg) was prescribed to each patient at random. All patients were encouraged to follow the prescribed diet during the study period, and compliance on therapy intake was checked via a questionnaire during the final visit (self-reported questionnaire that assesses adherence to medication refills and taking behavior). The control group consisted of patients who did not agree to receive supplements but continued their diet-only intervention. After 8 weeks of treatment, patients were reassessed (visit 2), and the study was ended.

### 2.2. Measurements

During the study period, blood samples were taken twice: at the beginning of the study (visit 1) and at the end of the study (follow-up visit 2) to assess the serum levels of the total cholesterol (TC), LDL-C, and high-density lipoprotein cholesterol (HDL-C), triglycerides (TG), aspartate aminotransferase (SGOT), alanine aminotransferase (SGPT), and creatine phosphokinase (CPK). Moreover, in visit 2, height and weight were measured, and body mass index (BMI) was calculated as the ratio of weight (in kilograms) to squared height (in meters). The serum lipids (TC, TG, LDL-C, and HDL-C) were measured with an enzymatic colorimetric assay (Dimension Vista 500 System, Siemens, Munich, Germany). Liver enzymes consisting of alanine aminotransferase (SGPT), aspartate aminotransferase (SGOT), gamma-glutamyl transferase (γ-gt), and creatine kinase (CPK) were measured using an enzymatic method (Dimension Vista^®^ 500 System Siemens, Munich, Germany).

The study was conducted in accordance with the Declaration of Helsinki and was approved by the Institutional Review Board of the Athens Medical Center General Hospital, Athens, Greece (IRB protocol:01122022/100).

## 3. Statistical Analysis

The normality of the distribution of our data was analyzed using the Kolmogorov– Smirnov test. Differences of data with skewed distributions were analyzed with the Wilcoxon nonparametric test. The Kruskal–Wallis test was used to examine significant differences between groups with respect to dependent variables. A Dunn–Bonferroni test was then used to compare the groups in pairs to investigate which was significantly different. The results were presented as the mean values ± standard deviation (SD). Moreover, the difference between time points was indicated in percentage. Statistical Package of Social Sciences 21.0 (SPSS Inc., Chicago, IL, USA) was used for the statistical analysis, and *p*-values < 0.05 were deemed significant.

## 4. Results

One hundred and seventy-eight subjects with hypercholesterolemia were screened, of which 122 subjects met our eligibility criteria. After an 8-week lifestyle modification period, 105 patients were evaluated during visit 1 and randomized to receive a compound containing 10 mg of monacolin (Group A, *n* = 39), a compound containing 3 mg of monacolin (Group B, *n =* 33), and the rest were advised to continue the diet without any supplement (Control group, *n =* 33). Of those, eight patients (six in the control group and two in Group B) withdrew from the study for reasons illustrated in Figure 2. No severe adverse events occurred during the study. One patient required treatment interruption due to dizziness and reduced appetite during the first 3 days of therapy. The remaining 97 patients who completed the study were included in the analysis (visit 2).

The mean age of the overall population was 56.7 ± 10.0 years (range 40–80 years, 13 males/84 females). Table 2 shows the main characteristics of the study population at the baseline. There were no differences among the three groups in the baseline demographics and laboratory findings. There was a significant difference between the three groups after treatment with respect to all of the lipid parameters (Table 3). Comparisons of the lipid changes with the therapy between the three groups (means and means of percentile) are illustrated in Table 4 and Table 5, respectively. After treatment, LDL-C decreased by 26.46% on average (*p* < 0.001) during treatment with 10 mg of monacolin, by 16.77% on average during treatment with 3 mg of monacolin (*p* < 0.001), and increased by 1.6% on average (ns) in the diet-only group (Figure 3). Regarding TG concentrations, we observed a slight but significant reduction with treatment-only in the high-dose-treated group (mean −4.25%; 95% CI of mean −11.11 to 2.61).

## 5. Discussion

A significant proportion of the population has increased LDL-C concentrations without additional cardiovascular risk factors and, thus, are categorized as low-to-moderate CV risk. These patients are typically not candidates for lipid-lowering therapies and are instead advised to make lifestyle modifications, such as dietary adjustments and exercise. As recently recommended, subjects with CV risk who do not have a LDL-C goal but who are not eligible for statin treatment (or who are unable to take them) may consider treatment with nutraceuticals containing lipid-lowering substances such us phytosterols and RYR [3].

A meta-analysis of 20 randomized control studies with 6653 participants (follow-up between 2 months and 3.5 years) and a monacolin K dose varying from 4.8 to 24 mg per day showed that RYR supplementation lowered LDL-C compared to a placebo (−39.44 mg/L) [16]. However, since monacolin K is structurally identical to lovastatin, the same safety concerns related to some side effects commonly associated with statins are present with that compound as well [17,18].

Our study investigated the effects of supplementation with low and high daily doses of RYR on the absolute changes of LDL cholesterol in a Greek population with mild dyslipidemia without known cardiovascular disorders. The main finding of this randomized controlled study is the evidence that a lower daily dose of RYR, containing 3 mg of monacolin K, can reduce the LDL-C levels by approximately 17% without any serious adverse effects. A study conducted by Heinz T. et al. with a similar study protocol investigated the effect of a low daily dose of 3 mg monacolin K from RYR on the concentration of LDL-C in a randomized, placebo-controlled intervention [19]. The study included 80 participants with mild-to-moderate hypercholesterolemia who were randomly assigned to either the treatment group, receiving 3 mg monacolin K, or the placebo group for a period of 12 weeks. In line with our findings, the results of the study showed a statistically significant reduction in LDL-C concentration in the treatment group compared to the placebo group. Specifically, the mean reduction in LDL-C concentration was 20.4% in the treatment group and 3.3% in the placebo group [19]. It is worth noting that, in the majority of available studies [20,21], monacolin was administered to statin-intolerant patients with mild-to-moderate hypercholesterolemia (LDL < 180 mg/dL). While the LDL reduction observed in these patients is notable, it remains uncertain whether this translates into an overall cardiovascular risk benefit. Furthermore, the effectiveness of monacolin-based nutraceuticals in patients with more severe hyperlipidemia has not been validated, raising questions about the appropriate timing for initiating such supplements.

Combining RYR with natural products that have different mechanisms of action may result in a synergistic effect [13,22]. For instance, while RYR can inhibit the HMG-CoA reductase enzyme, its potential may be further enhanced when paired with other nutraceuticals, such as plant sterols, which increase lipid excretion in the bowels. On the other hand, several cases with severe side effects have been reported [13] with RYR-based natural products, predominately in patients with a previous intolerance for statins. The remaining ingredients of the two supplements studied in our cohort were identical, indicating that the difference in LDL-C reduction was mainly attributed to different monacolin doses.

In Mediterranean countries, olive leaf extract (OLE) has been traditionally used as an herbal remedy [23] due to its high content of phenolic antioxidant, particularly oleuropein. OLE has been found to contain more phenolic antioxidant than olive fruit or olive oil [24,25] and also possesses lipid-lowering properties [26,27]. Although the mechanism of action of OLE’s health benefits is not fully understood, animal studies suggest that it may reduce cholesterol synthesis in rat hepatocytes by decreasing the activity of hydroxymethylglutaryl-CoA reductase [27]. A randomized, double-blind trial without a control group reported positive effects of a phenolic-rich OLE on blood pressure, as well as reductions in plasma total cholesterol, LDL-cholesterol, and triglyceride levels [28].

The antioxidant, lipid-lowering, and antiatherogenic properties of polyphenols and flavonoids present in grape seed extract (GSE) have also attracted interest in research studies [29]. GSE is rich in unsaturated fatty acids such as linoleic acid, linolenic acid, oleic acid, and palmitic acid, which have been shown to lower total cholesterol and LDL-cholesterol levels [30]. However, the effect of GSE on lipid profile is not yet conclusive, as some studies have reported a significant decrease in LDL-cholesterol and triglyceride levels but no effect on the total cholesterol and HDL-cholesterol levels [31]. Multiple trials have been conducted to assess the efficacy of grapes and their products on the levels of liver enzymes with controversial results. In particular, while some studies demonstrated beneficial effects [32], others found no benefits [33]. Although the exact mechanism of grape products on the liver function is still unclear, some plausible biological mechanisms may explain the beneficial effects of grape products on liver enzymes. Grape products have sown antioxidant and anti-inflammatory properties and thereby may have protective effects against liver abnormalities [34,35] due to their resveratrol content by reducing inflammation, inhibiting cellular stress, suppressing inflammatory gene expression, and enhancing peroxisome receptor-proliferative activity [36,37]. Thus, whether GSE played a protective role, and if so, to what extent, in the absence of the development of SAMS syndrome in our cohort remains questionable.

We understand that our study has some limitations. The major limitation of the present study is the short study period. The 8-week study period may not be adequate to evaluate the long-term efficacy of supplements and may underestimate dosage-dependent differences. Second, the sex ratio in our cohort was not representative of the general hypercholesteremic population, since participants were predominantly women; of note is that men reach the ASCVD risk threshold to receive lipid-lowering therapy for primary prevention more commonly than women, given that the male gender is a nonmodifiable risk factor for ASCVD events. Studies suggest that there has been a historical underrepresentation of women in clinical trials investigating statin therapy, which has led to some uncertainty about the benefits of statins in women, particularly for those without established cardiovascular disease [21,38]. Therefore, strategies aiming to lower LDL-C concentrations when statins are not mandated by the risk estimation could be of greater use for women overall. Nevertheless, our results cannot be generalized, since the number of male participants was restricted. Third, we did not treat with a placebo the patients forming the control group, and compliance to lifestyle modifications may be biased. Plant-based diets that are rich in vegetables, fruits, whole grains, nuts, and olive oil, such as the Mediterranean diet, have been shown to reduce inflammation and lower the risk of cardiovascular disease [39]. Clark et al. conducted a study to investigate the effect of a Mediterranean diet intervention on cardiometabolic outcomes in healthy older adults [40]. The study demonstrated that the intervention, which lasted for 6 months, significantly reduced the dietary inflammatory index (DII) scores compared to a habitual Australian diet. While the diet administered to our patients was for a shorter period of time, it is unclear what the cumulative effect of the diet was on the overall treatment results. Measurement of the CoQ10 serum levels of the study participants could therefore minimize the bias of our results, particularly in terms of evaluating the diet effect, but this was not feasible in our cohort. Although there is evidence to suggest that grape seed extract and olive leaf extract have health benefits [26,27,28,29], the precise mechanisms by which they exert these effects are not fully understood. Further research should investigate the biological pathways involved, which could lead to the development of more effective neutrochemical combinations. Obviously, studies with a larger control group and a longer follow-up period are needed to validate our promising preliminary results to determine whether the supplement has any cumulative effects, whether there are any adverse events that only emerge over time, and whether there may be some interactions with medications or other supplements that have not yet been identified.

## Figures and Tables

**Figure 1 nutrients-15-02682-f001:**
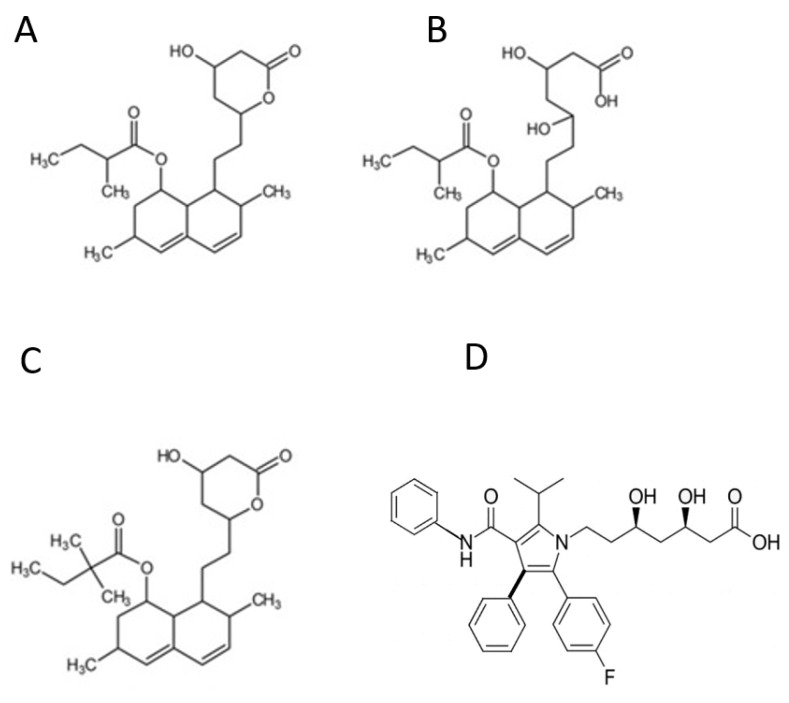
Chemical structures of monacolin K and the mainly used statins. Footnote: Monacolin K (Lovastatin) may occur in red yeast rice as lactone (**A**) or hydroxy acid (**B**). Simvastatin (**C**). Atrovastatin (**D**).

**Figure 2 nutrients-15-02682-f002:**
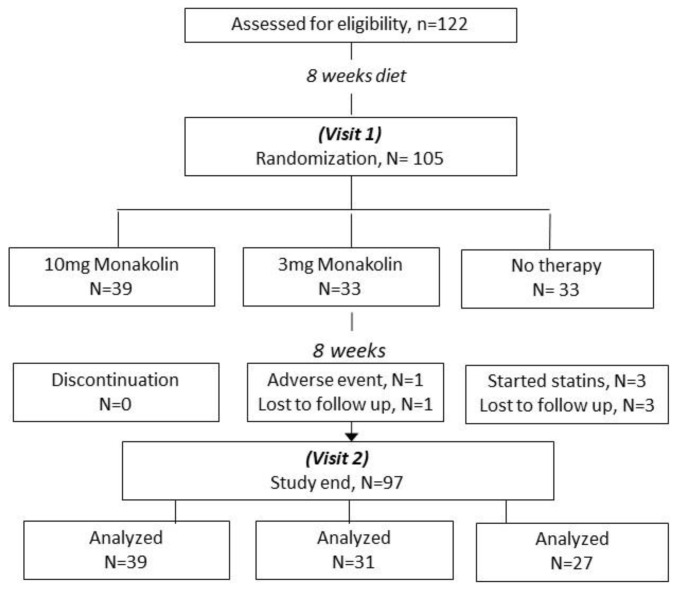
Study chart flow.

**Figure 3 nutrients-15-02682-f003:**
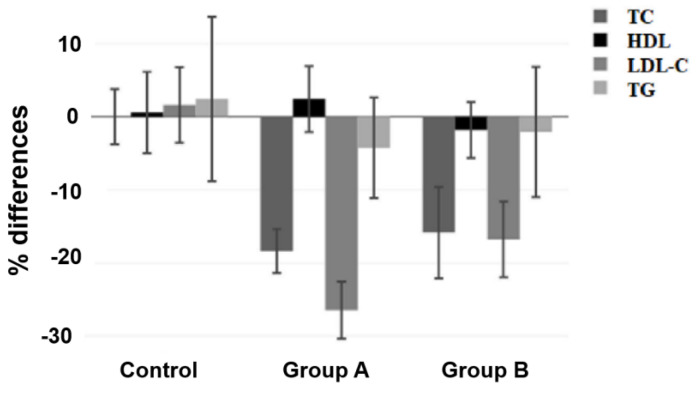
Percentage of differences in the lipid parameters after treatment. Data presented as means. Error bars indicate upper and lower 95% coefficient intervals of the means.

**Table 1 nutrients-15-02682-t001:** Chemical composition of the supplement.

Red Yeast Rice Monacolin K	10 mg or 3 mg
CoQ10	2 mg
Vitamin B5	6 mg
Vitamin B6	1.4 mg
Vitamin B2	1.4 mg
Vitamin B1	1.1 mg
Folic acid	200 mcg
Biotin	50 mcg
Vitamin B12	2 mcg
Olive leaf extract	50 mg
Grape seed extract	50 mg

Footnote: Ingredients per caps (daily dose).

**Table 2 nutrients-15-02682-t002:** Baseline characteristics of the study population.

Variable	Group A *n* = 39	Group B *n* = 31	Control *n* = 27	*p*
Age	56.89 ± 9.12	55.83 ± 9.25	57.42 ± 12.29	0.744
Gender female, *n* (%)	34 (87.2)	27 (87.1)	23 (85.2)	0.812 *
BMI (kg/m^2^)	25.39 ± 3.23	24.76 ± 3.53	27.10 ± 3.31	0.052
TC (mg/dL)	259.8 ± 25.9	255.3 ± 25.1	247.0 ± 21.9	0.174
LDL-C (mg/dL)	167.7 ± 14.22	166.5 ± 14.31	162.2 ± 15.42	0.281
HDL (mg/dL)	64.12 ± 16.79	62.16 ± 14.69	56.65 ± 11.44	0.191
TG (mg/dL)	125.9 ± 73.5	132.3 ± 56.1	146.6 ± 51.4	0.062
SGOT (U/L)	26.71 ± 6.04	23.89 ± 6.52	24.52 ± 11.46	0.103
SGPT (U/L)	29.71 ± 6.20	25.53 ± 7.96	27.68 ± 13.37	0.102
γ-GT (U/L)	20.94 ± 6.88	17.56 ± 9.39	18.50 ± 10.66	0.105
CPK (U/L)	79.33 ± 25.17	88.42 ± 40.08	73.69 ± 30.74	0.165

Footnote: TC: total cholesterol; LDL-C: low-density lipoprotein cholesterol; HDL: high-density lipoprotein cholesterol; TR: triglycerides; Sgpt: alanine transaminase; Sgot: aspartate transaminase; γ-gt: gamma-glutamyl transferase; CPK: creatine phosphokinase. Group A: 10 mg of monacolin; Group B: 3 mg of monacolin. *p*: one-way analysis of variance; *p* *: x^2^ test of independence. All values are expressed as the mean ± standard deviation.

**Table 3 nutrients-15-02682-t003:** Biochemical profiles before and after treatment.

Variable	Group A *n* = 39			Group B *n* = 31			Control *n* = 27			
	Baseline	Study End	*p*	Baseline	Study End	*p*	Baseline	Study End	*p*	*p*
BMI (kg/m^2^)	25.39 ± 3.23	25.28 ± 3.47	0.456	24.76 ± 3.53	23.57 ± 5.53	0.749	27.10 ± 3.31	25.95 ± 6.23	0.222	0.175
TC (mg/dL)	259.8 ± 25.9	211.49 ± 28.7	<0.001	255.3 ± 25.1	198.58 ± 71.3	<0.001	247.0 ± 21.9	238.93 ± 61.93	0.657	0.002 ^a^
LDL-C (mg/dL)	167.7 ± 14.22	123.38 ± 23.74	<0.001	166.5 ± 14.31	118.97 ± 44.2	<0.001	162.2 ± 15.42	157.41 ± 35.07	0.722	<0.001 ^b^
HDL (mg/dL)	64.12 ± 16.79	65.08 ± 16.53	0.227	62.16 ± 14.69	54.23 ± 20.79	0.198	56.65 ± 11.44	54.19 ± 15.12	0.482	0.026 ^c^
TG (mg/dL)	125.9 ± 73.5	116.77 ± 55.22	0.004	132.3 ± 56.1	112.42 ± 68.34	0.071	146.6 ± 51.4	138.56 ± 51.18	0.424	0.040 ^d^
SGOT (U/L)	26.71 ± 6.04	28.18 ± 7.1	0.349	23.89 ± 6.52	22.52 ± 11.46	0.508	24.52 ± 11.46	24.19 ± 9.75	0.712	0.188
SGPT (U/L)	29.71 ± 6.20	29.05 ± 7.85	0.289	25.53 ± 7.96	21.81 ± 12.08	0.124	27.68 ± 13.37	24.81 ± 14.58	0.492	0.150
γ-GT (U/L)	20.94 ± 6.88	26.21 ± 7.33	<0.001	17.56 ± 9.39	19.23 ± 11.68	0.170	18.50 ± 10.66	19.67 ± 13.09	0.644	0.01
CPK (U/L)	79.33 ± 25.17	77.92 ± 24.89	0.898	88.42 ± 40.08	74.29 ± 48.05	0.509	73.69 ± 30.74	71.59 ± 37.58	0.799	0.379

Footnote: TC: total cholesterol; LDL-C: low-density lipoprotein cholesterol; HDL: high-density lipoprotein cholesterol; TR: triglycerides; Sgpt: alanine transaminase; Sgot: aspartate transaminase; γ-gt: gamma-glutamyl transferase; CPK: creatine phosphokinase; Group A: 10 mg of monacolin; Group B: 3 mg of monacolin; *p*: Nonparametric Wilcoxon test for paired values; *p*: Kruskal–Wallis test and Dunn’s multiple comparisons test between groups (mean rank difference, adjusted *p*-value). ^a^ Group A vs. Group B: −2.37, *p* = ns; Group A vs. Control: −23.76, *p* = 0.002; Group B vs. Control: −21.39, *p* = 0.012. ^b^ Group A vs. Group B: −2.93, *p* = ns; Group A vs. Control: −38.27, *p* < 0.001; Group B vs. Control: −35.35, *p* < 0.001. ^c^ Group A vs. Group B: 12.5, *p* = ns; Group A vs. Control: 18.16, *p* = 0.03; Group B vs. Control: 5.66, *p* = ns. ^d^ Group A vs. Group B: −0.12, *p* = ns; Group A vs. Control: −16.22, *p* = 0.021; Group B vs. Control: −16.1, *p* = 0.03.

**Table 4 nutrients-15-02682-t004:** Comparison of the mean lipid alterations after therapy.

		Treatment			
		Group A	Group B	Control	*p*
TC	Mean	−48.38	−41.58	1.07	<0.001 ^a^
	95% (CI of Mean)	−56.76; −40.01	−59.12; −24.05	−7.96; 10.11	
	SD	26.69	49.81	23.95	
	Min	−119	−278	−51	
	Max	2	19	52	
LDL-C	Mean	−44.38	−30.19	1.19	<0.001 ^b^
	SD	21.64	22.29	21.12	
	Min	−103	−75	−37	
	Max	−7	2	45	
	95% (CI of Mean)	−51.18; −37.59	−38.04; −22.35	−6.78; 9.15	
HDL	Mean	0.95	−2.13	−0.37	0.229 ^c^
	SD	9.93	8.61	7.03	
	Min	−23	−25	−12	
	Max	37	9	20	
	95% (CI of Mean)	−2.17; 4.06	−5.16; 0.9	−3.02; 2.28	
TG	Mean	−11.77	−4.65	−2.63	0.768 ^d^
	SD	27.79	36.63	36.5	
	Min	−110	−93	−61	
	Max	34	97	111	
	95% (CI of Mean)	−20.49; −3.05	−17.54; 8.25	−16.4; 11.14	

Footnote: *p*: Kruskal–Wallis test. Dunn’s multiple comparisons test between groups (mean rank difference, adjusted *p*-value). ^a^ Group A vs. Group B: −11.51, *p* = ns; Group A vs. Control: −43.02, *p* = <0.001; Group B vs. Control: −31.52, *p* = 0.001. ^b^ Group A vs. Group B: −14.33, *p* = ns; Group A vs. Control: −44.99, *p* < 0.001; Group B vs. Control: −30.66, *p* < 0.001. ^c^ Group A vs. Group B: 9.31, *p* = ns; Group A vs. Control: 10.64, *p* = ns; Group B vs. Control: 1.33, *p* = ns. ^d^ Group A vs. Group B: −3.9, *p* = ns; Group A vs. Control: −4.55, *p* = ns; Group B vs. Control: −0.65. *p* = ns.

**Table 5 nutrients-15-02682-t005:** Percentage of the changes in the lipid levels.

Variable	Percentage of Difference, %	Treatment			
		Group A	Group B	Control	*p*
TC	Mean	−18.37	−15.85	0	<0.001 ^a^
	SD	9.59	17.75	10.01	
	Min	−42.31	−100	−23.23	
	Max	0.81	6.91	19.7	
	95% (CI of Mean)	−21.38; −15.36	−22.1; −9.6	−3.78; 3.78	
LDL-C	Mean	−26.46	−16.77	1.61	<0.001 ^b^
	SD	12.45	14.68	13.75	
	Min	−58.86	−41.67	−22.84	
	Max	−4.12	20.35	31.03	
	95% (CI of Mean)	−30.37; −22.55	−21.93; −11.6	−3.58; 6.79	
HDL	Mean	2.43	−1.82	0.58	0.256 ^c^
	S	14.32	10.87	14.78	
	Min	−28.05	−28.05	−18.52	
	Max	55.22	16.07	54.05	
	95% (CI of Mean)	−2.07; 6.92	−5.64; 2.01	−4.99; 6.16	
TG	Mean	−4.25	−2.09	2.42	0.743 ^d^
	SD	21.85	25.29	29.81	
	Min	−43.81	−37.37	−47.62	
	Max	59.65	56.73	106.73	
	95% (CI of Mean)	−11.11; 2.61	−10.99; 6.82	−8.82; 13.67	

Footnote: *p*: Kruskal–Wallis test. Dunn’s multiple comparisons test between groups (mean rank difference, adjusted *p*-value). ^a^ Group A vs. Group B: −11.55, *p* = ns; Group A vs. Control: −41.78, *p* = <0.001; Group B vs. Control: −30.24, *p* = 0.001. ^b^ Group A vs. Group B: −14.69, *p* = ns; Group A vs. Control: −45.77, *p* < 0.001; Group B vs. Control: −31.08, *p* < 0.001. ^c^ Group A vs. Group B: 9.12, *p* = ns; Group A vs. Control: 10.12, *p* = ns; Group B vs. Control: 1, *p* = ns. ^d^ Group A vs. Group B: −1.12, *p* = ns; Group A vs. Control: −5.3, *p* = ns; Group B vs. Control: −4.17, *p* = ns.

## Data Availability

The datasets used and analyzed during the current study are available from the corresponding author upon reasonable request.

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
