# Peer review of "Low Dose Monacolin K Combined with Coenzyme Q10, Grape Seed, and Olive Leaf Extracts Lowers LDL Cholesterol in Patients with Mild Dyslipidemia: A Multicenter, Randomized Controlled Trial"

_nutrients, 2023, doi:10.3390/nu15122682_

Round 1

Reviewer 1 Report

Angelopoulos et al. reported an interesting trial, investigating a novel combination of monacolin K and coenzyme Q10. The authors well summarized the main purpose of the study, including the rationale. They also discussed the limitations of the study.

However, I have several questions that the authors should address to make the paper more complete and insightful.

Major concerns

-          I believe that the discussion should be implemented with more references and highlight more potential impact on clinical practice.

-          Regarding olive leaf extract (OLE), grape seed extract (GSE), I am wondering what are their specific nutraceutical properties and what is the exact rationale for combining them with RYR and CoQ? Even though the authors mentioned their previous paper, I believe that this is a key point that could be added to the discussion and highlight the novelty of this combination.

-          Could the authors include or indicate  any potential sources of bias or confounding factors that may have influenced the results?

-          Could the authors better define or suggest areas for future research?

- Could the authors better describe the control patients used in this study, if any?

Minor concerns

-          I would suggest the authors add an abbreviation list.

-          Could the authors mention more information about, on average, the Mediterranean diet could contribute to CoQ intake in these patients in comparison to other diets and exert antilipidemic effects? Taking into account that Mediterranean Diet is full of CoQ-rich food such as several wholegrains and nuts, I am wondering if it could have an impact. For example, could be mentioned  "Clark et al. Adherence to a Mediterranean Diet for 6 Months Improves the Dietary Inflammatory Index in a Western Population: Results from the MedLey Study" or other study/ analysis performed.

-          Do you know, if any, how much CoQ is in olive leaf extract (OLE), E, and grape seed extract (GSE)?

-          Could the authors better describe the study design and methods used to recruit participants?

-          After how many weeks  lipid profile was assessed (102) and when exactly visit 1 and visit 2 were performed (120)?

-          Do the author think that heterogeneity of the study population could heavily have affected the findings described?

-          Could the authors describe better the questionnaire administered to the patients, mentioned in 108?

-          Do the authors have any chance to measure CoQ levels in the patients (plasma sample for example)?

-          In the discussion, the results should be interpreted in light of the current literature or also in comparison to previous studies such as “Heinz T,et al. Low daily dose of 3 mg monacolin K from RYR reduces the concentration of LDL-C in a randomized, placebo-controlled intervention.”

-          The author mentioned (213-214) RYR has a hypocholesterolemic effect for inhibiting HMG-CoA, acting similarly to statins. I believed that this part sounds redundant in the discussion and should be expanded in the introduction. Thus, the authors could take chance to describe more about this pivotal property of RYR. Could the authors also show a short graphical abstract showing the structure of Monacolin K in comparison to well-known statins?

-          Do the authors believe or have stronger references about possible lifestyle modifications that could affect the lowering of lipids?

-          Do the authors believe there could be a specific time, before the onset of severe dyslipidemia, when these nutraceutical interventions should be applied and have beneficial effects?

Reviewer 2 Report

The article entitled " Low dose monacolin K combined with coenzyme Q10, Grape 2 

Seed, and Olive Leaf Extracts lowers LDL cholesterol in patients with mild dyslipidemia: A multicenter, randomized con- 4 trolled trialrefers to the pharmacological applications of the monacolin Kwhich has been found a valid alternative to statin therapy. The article is interesting and well exposing

However, there are some issues:

• The study has a significant limitation due to the small number of control groups. Increasing the number of control groups would enhance the statistical reliability and strengthen the credibility of the findings. • In addition to specifying the dosage of the therapy, it would be beneficial to also provide information on the duration and frequency of patients' monacolin K intakein the “Material and Methods” section, This data could be valuable in the context of replacement therapy. • In the discussion, line 288, not is clear the interaction and synergistic effect 

between RYR and coezyme Q10. 

• In line 240, why this therapy could be more useful for women? • The overall discussion lacks clear and well-supported arguments, and would benefit from additional emphasis. Furthermore, the conclusions drawn from the discussion should be explicitly stated and supported with evidence.
